# Gamli – Icelandic Oral History Corpus:
# Design, Collection and Evaluation

**Luke O'Brien**
Tiro
luke@tiro.is

**Finnur Ágúst Ingimundarson**
The Árni Magnússon Institute
for Icelandic Studies
fai@hi.is

**Jón Guðnason**
University
of Reykjavik
jg@ru.is

**Steinþór Steingrímsson**
The Árni Magnússon Institute
for Icelandic Studies
steinthor.steingrimsson
@arnastofnun.is

## Abstract

We present *Gamli*, an ASR corpus for Icelandic oral histories, the first of its kind for this language, derived from the Ísmús ethnographic collection. Corpora for oral histories differ in various ways from corpora for general ASR, they contain spontaneous speech, multiple speakers per channel, noisy environments, the effects of historic recording equipment, and typically a large proportion of elderly speakers. Gamli contains 146 hours of aligned speech and transcripts, split into a training set and a test set. We describe our approach for creating the transcripts, through both OCR of previous transcripts and post-editing of ASR output. We also describe our approach for aligning, segmenting, and filtering the corpus and finally training a Kaldi ASR system, which achieves 22.4% word error rate (WER) on the Gamli test set, a substantial improvement from 58.4% word error rate from a baseline general ASR system for Icelandic.

## 1 Introduction

Icelandic open-licensed speech corpora have in recent years grown in volume and numbers, there are now *Talrómur* (Sigurgeirsson et al., 2021), *Málrómur* (Steingrímsson et al., 2017), *Samrómur* (Mollberg et al., 2020) and the Althingi's Parliamentary Speeches corpus (Helgadóttir et al., 2017; Nikulásdóttir et al., 2018) to name a few. However both historical speech and older speakers are underrepresented in these corpora. For instance, regarding older speakers, in Samrómur, the largest open-licensed ASR corpus for Icelandic (2233 hours in the latest release, Hedström et al. 2022), only 4.8% of speakers are over 60 years old.

*Gamli*, the oral history speech corpus presented in this paper differs from that in many ways. Firstly, it contains, predominantly, spontaneous speech in the form of interviews, secondly, it has a very high ratio of older speakers (94.8% of speakers are over 60 years old), thirdly, background noise is common as well as noise artefacts from historical recording equipment and lastly, historic dialects (word choice and accent) are much more prevalent than in existing corpora.

The corpus contains 146 hours of aligned speech and transcripts split into a training set and a test set. This data, based on valuable historical 20th century recordings stored at the Department of Ethnology and Folklore at The Árni Magnússon Institute for Icelandic Studies, is therefore an important addition to the existing Icelandic speech corpora.[1]

The custom ASR system presented in this paper along with the corpus will in due course be used to automatically transcribe all of the ethnographic audio recordings stored at the institute. The transcripts will then be made available on the online portal *Ísmús*[2] and paired with the respective recording.

## 2 Related Work

For many years, ASR systems have been trained on unaligned transcriptions (Panayotov et al., 2015) and even approximate transcriptions of spontaneous speech (Jang and Hauptmann, 1999). In the case of Icelandic ASR for spontaneous speech, there has been an ongoing project (Helgadóttir et al., 2017), (Helgadóttir et al., 2017) to align and filter Icelandic parliamentary transcripts for ASR in order to reduce the manual work involved in transcribing parliamentary proceedings.

---

[1]The corpus is available under an open license at http://hdl.handle.net/20.500.12537/310

[2]www.ismus.is

Creating the corpora involves text normalization, time-alignment, and filtering utterances.

While ASR for oral histories is new for Icelandic, it is already being used in other languages. For example, the first large project was the MALACH project (Psutka et al., 2002) in 2002, where ASR transcriptions were used for indexing oral history archives and making them more searchable. However, some authors still consider oral history speech recognition an open problem (Picheny et al., 2019; Gref et al., 2020) and a recent study (Gref et al., 2022) found that human word error rate was 8.7% on a German oral history corpus (taking into account case-sensitivity and annotation of hesitations). Whereas Lippmann (1997) found a human word error rate of less than 4% on the Switchboard corpus of spontaneous telephony speech and less than 0.4% on the Wall Street Journal corpus of clear read speech. This suggests that the minimum possible word error rate for ASR might be much higher on oral histories than it is for cleaner speech corpora.

One other factor that makes oral history ASR an interesting challenge is the particularly high ratio of older speakers. It has been noted by Vipperla et al. (2008) that for general ASR models, WER correlates strongly with age, even throughout a single speakers lifetime. This could be caused by multiple changes in aging voices, such as slower speaking rate, changes in F0 (decrease for males and increase for females), increase in jitter and shimmer (all from Vipperla et al. (2008)), some of which could be mitigated by increasing the number of older speakers in the training set. However, other changes might not be so easily solved, such as a reduction of tongue and jaw strength and an increase in breathiness (all from Vipperla et al. (2008)) which could reduce articulatory precision.

There are three main use-cases for oral history speech recognition. First, to index oral archives for spoken document retrieval. Second, to provide transcripts to aid listeners. Third, as a hypothesis transcript for post-editing. For each of these use-cases, it's important to determine the minimum acceptable ASR performance. For the first use-case, indexing, Chelba et al. (2008) found that using ASR output significantly improves spoken document retrieval performance compared to only using the accompanying text meta data, even when WER is as high as 50%. More recently, Fan-Jiang et al. (2020) used a BERT-based retrieval model with query reformulation and managed to get impressive results for document retrieval of Mandarin news when using erroneous recognition transcripts (35%). The accuracy was 0.594 with the erroneous transcripts and 0.597 with reference transcripts. This suggests that for indexing, acceptable WER may be even higher than 35%. For the second use-case, to provide transcriptions as an aid to listeners, Munteanu et al. (2006) found that transcripts with a 25% WER improved listeners' understanding more than listening to audio without a transcript, however they found that understanding was reduced when the transcripts had 45% WER, suggesting that a maximum acceptable WER is somewhere between 25% and 45%. For the third use-case, post-editing, Gaur et al. (2016) found that for recordings of Ted Talks, ASR transcriptions with less than 30% WER sped up the transcription process but if the WER was higher than 30% it slowed transcribers down.

## 3 Origin of the corpus

The ethnography collection of the Department of Ethnology and Folklore at The Árni Magnússon Institute for Icelandic Studies contains more than 2,300 hours of audio recordings of oral heritage and traditions, with a little less than 2,500 interviewees. The oldest material are recordings made on wax cylinders in the early 20th century and the collection is continually expanding with new material being added every year.

The bulk of the collection, however, consists of recordings from the 1960's and 1970's, mainly the work of three collectors. Their focus was to gather ethnographic material from all of Iceland, first and foremost from older generations — the majority of the informants were born before or around the turn of the 20th century,

This resulted in an extensive collection of legends and fairy tales, accounts of beliefs and customs, poems, hymns, nursery rhymes, Icelandic ballads (*rímur*), occasional verses and more, with the material being variously spoken, sung or chanted. Apart from recited verse and that which is sung or chanted the speech is spontaneous. Accompanying the recordings is detailed metadata on the speaker, time and location of recording, as well as various other parameters such as genre (for different kinds of verse or prose material, e.g. poems or nursery rhymes, fairy tales or legends etc.), mode of performance (sung, chanted, spoken), key

words, content (short summary, description), tale-types and motifs (in folktales and legends).

### 3.1 Speaker distribution in the collection

In their work the collectors mainly relied on a snowball method of sorts, asking speakers to point them to other possible informants, as well as contacting teachers or clergy to enquire about interesting subjects in their region. Speaker profession is often listed in the metadata and most of the speakers were workers, farmers, fishermen, housewives etc., with little formal education.

Gender was probably not a decisive factor at the outset and the total ratio is 57.6% male speakers and 42.4% female, i.e. based on the number of speakers. However, if audio length for each gender is included the difference increases quite a bit, i.e. 1504 hours (65%) for men vs. 821 hours (35%) for women.

As mentioned, the data in the collection also stands out in that that the age of the speakers is higher than in other existing Icelandic corpora. The oldest speaker in the collection was 105 years old at the time of recording in 1954 and the oldest speaker in the collection, with regards to date of birth, was born in 1827, and recorded in 1904 (not included in the *Gamli* corpus). In fact, 72.4% of the speakers are older than 63 and 31.4% are 71-80 years old. In *Gamli* this ratio is substantially higher, as detailed in Section 4.

### 3.2 Regional features in pronunciation

The speakers in the collection are from all over Iceland and therefore reflect the various regional differences in pronunciation much better than recently recorded speech corpora such as *Samró-mur*, due to the fact that these regional features either have already more or less disappeared or are gradually disappearing. Amongst these features is for example the "hard" pronunciation of /p, t, k/ (still a distinct feature) and voiced pronunciation of /l, m, n/ before /p, t, k/ in North-Iceland, *rn-*, *rl*-pronunciation in South-East-Iceland, monophthongs before /ng, nk/ in the North-West etc.

While these features are not tagged in any way in the *Gamli* corpus, the ASR system trained on the corpus seems to work well on these features, with possibly the exception of labial or velar stops before [ð], such as [hapðɪ] instead of [havðɪ] for *hafði* or [lakðɪ] instead of [laɣðɪ] for *lagði*. We have, however, not inspected this systematically,

so it needs further looking into to state the precision with any certainty.

### 3.3 Recording procedure

Most of the recordings were made at the speakers' homes, in many cases in elderly homes, and carried out by the interviewer. It was not uncommon that other people, e.g. children, spouses etc. were present during the recording sessions, but they were in most cases not meant to play a part in the recording. Because of this, and for various other reasons, some background noise and disturbances occur in the recordings, e.g. children playing, traffic sounds, phones ringing etc., but these are generally not prominent.

Much of the recordings were recorded using high quality reel-to-reel tape recording devices, although some were done by amateurs who weren't as well equipped, whereas a part of the recordings are from the recording studios of The Icelandic National Broadcasting Service (Þorsteinsdóttir, 2013).

The digitalization of these recordings began in the late 1990's and continued into the early 2000's with the recordings being converted into WAV format as well as compressed MP3s for online use.

## 4 Corpus content

*Gamli* contains 146 hours of transcribed audio broken down into

1. ∼ 111 hours from optical character recognition (OCR) of previous transcriptions in various formats

2. ∼ 35 hours of new transcriptions (post-edited from ASR output)

The 111 hours include 9 hours defined as a test set, which was manually reviewed and corrected and annotated with speaker ID and time alignments in the annotation tool *ELAN*. The test set contains recordings with 10 speakers, 5 women and 5 men, plus the interviewers (4 men) and serves for evaluating the system's performance.

A validation set has not been defined for the corpus as the acoustic model training in *Kaldi* (Povey et al., 2011) used a random sample of the training corpus for validation.

### 4.1 Speaker distribution in the corpus

The corpus contains 210 unique speakers, 90 women and 120 men (plus the interviewers: 14

| Data split | Hours | Male speakers | Female speakers | Total speakers |
|---|---|---|---|---|
| Training | 137 | 115 | 85 | 200 |
| Test | 9 | 5 | 5 | 10 |

Table 1: Data splits in Gamli

men and 1 woman). At the outset we aimed to have the gender ratio as equal as possible in the acoustic training data, but with three men surpassing 20 hours of speech each (with one topping at 29 hours) and accounting for more than one third of the entire data, that picture became quite distorted. As a result the gender bias in the corpus is even greater than in the collection itself, which is unfortunate, but simply reflects the data that was at hand, i.e. 73.5% vs. 26.5%, cf. Section 4.2.

The age ranges from 38 to 99, but most of the speakers are 60+ (94.8%), as shown in Figure 1, and the average age of the speakers is 77 years. This ratio is unprecedented in all existing corpora for Icelandic speech (cf. 4.8% in Samrómur as referred to in Section 1) and makes Gamli an important addition to that collection.

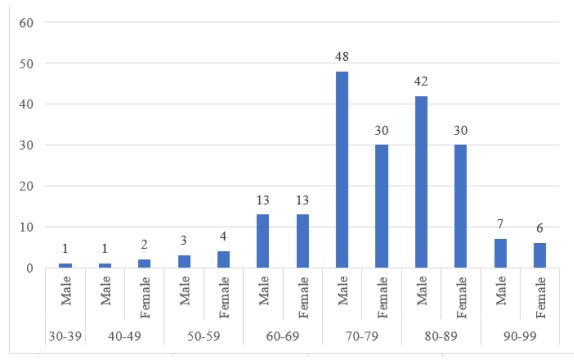

Figure 1: Age distribution of unique speakers in the training set

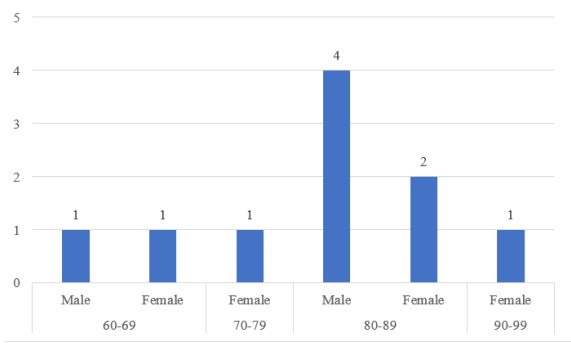

Figure 2: Age distribution of unique speakers in the test set

## 4.2 Corpus compilation

As mentioned, the largest part of the corpus, about 111 hours, stems from OCR of transcriptions at the Department of Ethnology and Folklore at The Árni Magnússon Institute for Icelandic Studies. These transcripts that were generated over several decades are not all in the same format (e.g. type-written, dot printed, printed Word documents) and therefore needed first to be processed, i.e. scanned and OCRed (the results of which varied depending on the format). These transcripts were then catalogued and paired with the respective recordings.

Once this ready data had been processed the first ASR output was produced and manually corrected. During that process it became evident that some of the recordings were ill suited at this stage as they often contained poetry, nursery rhymes and in some cases singing, where the ASR system could not be expected to do well as the focus was on spontaneous speech, where it performed much better (cf. Section 6).

As a result, we made use of the detailed metadata search parameters in the Ísmús portal in order to filter the best in-domain data for further training. We mainly relied on the so-called *form* parameter (*genre*) to try to exclude everything but spontaneous speech. This gave much better results and resulted in the 35 hours of post-edited data mentioned in Section 4.

## 4.3 Normalizing, aligning, segmenting and filtering the transcripts for ASR training

The transcripts in the training set did not have time alignments and some had OCR spelling errors. Therefore, we had to process the transcripts before using them to train the acoustic model. First, the text was normalized using the Regina normalizer developed in Sigurðardóttir (2021). Second, the text was aligned to the audio with Kaldi's segment long utterances function [3]. For this, a biased language model (based on the text) is combined with an existing acoustic model to force-align the audio, as detailed in section 2.2 of Manohar et al. (2017). It outputs aligned segments of less than 15 seconds each. Third, these segments are filtered with Kaldi's clean and segment data function [4] which again combines a biased language model

---

[3] https://github.com/kaldi-asr/kaldi/blob/master/egs/wsj/s5/steps/cleanup/segment_long_utterances_nnet3.sh

[4] https://github.com/kaldi-asr/kaldi/blob/master/egs/wsj/s5/steps/cleanup/

(based on the text) with an existing acoustic model and removes segments that were unintelligible to the decoder.

After filtering, 180 hours of interviews was reduced to 137 hours (24% reduction). However, much of this reduction can be attributed to silences in the audio, so to estimate the total amount of speech reduced, we note that the word count was reduced from 1,147,181 to 1,039422 (9.4% reduction).

Finally, after training an acoustic model on this in-domain data, the alignment, segmentation, and filtering was performed again. That final data constitutes the Gamli training set. The final model was then trained on that data.

## 5    Models (and out-of-domain data)

We trained a hybrid ASR system in Kaldi. That is, the language model and acoustic model were trained separately as opposed to an end-to-end system. For the acoustic and language models in the custom ASR system, we expanded the training sets with various out-of-domain data, which will be described in the following sections.

### 5.1    Acoustic Model

An acoustic model learns to map audio to a sequence of phonemes. The acoustic model is a TDNN (time-delayed neural network) chain model trained in Kaldi. It was trained on the in-domain data described above, but also on various out-of-domain data, which included the following datasets:

1. Althingi's Parliamentary Speeches.[5] A corpus of 514.5 hours of recorded speech from the Icelandic parliament (Helgadóttir et al., 2017)

2. 114.6 hours of speech from the first Samrómur release,[6] leaving out children.

3. 173.1 hours of unverified Samrómur data,[7] containing only speech with 50+ year old men and 60+ year old women.

4. 228.2 hours of the RÚV TV unknown speakers dataset.[8]

iVectors and MFCCs (Mel-frequency cepstral coefficients) are the inputs to the acoustic model. These are commonly used in Kaldi 'chain' models. The iVectors in particular are said to make the neural network speaker adaptive since the vectors themselves carry speaker identity information (Saon et al., 2013).

Data augmentation was also used to triple the entire training set. We added artificial noise and reverberation. For noisy data sets, e.g. call-center data sets, this is said to give better results than speed perturbations (Ko et al., 2017) and as was described earlier, background noise and disturbances are not uncommon in the data.

### 5.2    Language Model

A language model is necessary for outputting coherent texts, it learns a probability distribution for word sequences from a training corpus. The language modelling consists of a 3-gram language model for decoding and an RNN language model for rescoring. It was trained on the Gamli training set described in 4.2, as well as out-of-domain data. The out-of-domain data stems from the following sources:

1. The Icelandic Gigaword Corpus (IGC) (Steingrímsson et al., 2018). We use the sentences from the 2022 version of the IGC.[9]

2. Ethnographic data from the National Museum of Iceland in *Sarpur*.[10]

3. Audio file descriptions from Ísmús [11] for their content.

4. Place name data from the Icelandic Place Name Collection.[12]

### 5.3    Vocabulary and Pronunciation Dictionary

The pronunciation dictionary maps words to sequences of phonemes. For the vocabulary we used:

---

clean_and_segment_data_nnet3.sh

[5]Available at:   http://hdl.handle.net/20.500.12537/277

[6]Available at:   http://hdl.handle.net/20.500.12537/189

[7]Available at:   http://hdl.handle.net/20.500.12537/265

[8]Available at:   http://hdl.handle.net/20.500.12537/191

[9]http://hdl.handle.net/20.500.12537/254

[10]https://sarpur.is/

[11]https://ismus.is/

[12]nafnid.is

1. All the word forms from *The Database of Icelandic Morphology* (Bjarnadóttir et al., 2019).

2. OOV words from audio file descriptions in Ís-mús.

3. Vocabulary from the training set (only the data that was manually transcribed and not the OCR data); manually checked and added where appropriate.

4. OOV words from *Sarpur*; (manually checked and added where appropriate).

To get the phonemic transcriptions of each word a G2P model based on the *Icelandic Pronunciation Dictionary for Language Technology*[13] was used.

## 6 Evaluation

To assess the final ASR system's performance on the test set, we compare it to two baselines. The first is the out-of-domain system, which was trained in the same way as the final system but only on the out-of-domain data detailed in sections 5.1 and 5.2, not on the Gamli training set. The second baseline is the Samrómur "base" system [14]. This is a kaldi-trained system from a well-known dataset of read Icelandic speech, the acoustic mode is a TDNN chain model the language model is an n-gram model. While the ASR baseline systems achieved 36.7% and 58.4% respectively on the Gamli test set, the final ASR system performed better, achieving 22.4% WER on the same set, as shown in Table 2. This table compares the three overall systems, each including their own acoustic model and language model. However, it should be noted that the same lexicon and vocabulary were used for the final system and the out-of-domain system.

To investigate the differences in the systems, we also compare the performance when taking demographic information into account in Figure 3. As stated earlier, the test set contains 10 speakers and a total of 9 hours of audio.

To separate the effect that the Gamli training set had on acoustic model adaptation and language model adaptation, in Table 3, we compare WER when combining the out-of-domain models with

[13]Available at: http://hdl.handle.net/20.500.12537/99
[14]https://github.com/cadia-lvl/samromur-asr/tree/master/s5_base

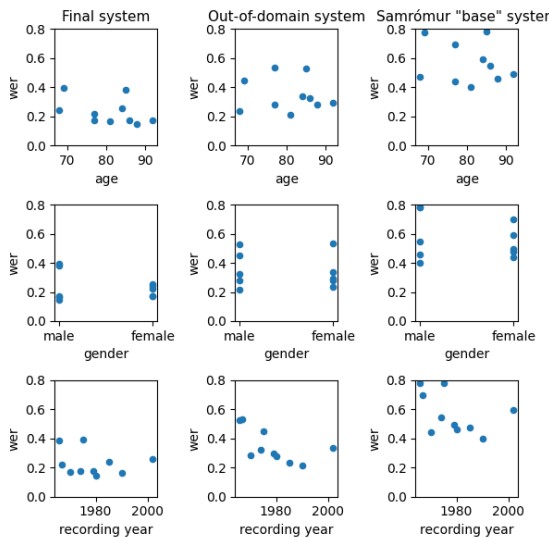

Figure 3: WER for the 10 unique speakers in the Gamli test set based on demographic information. Comparing the final system we trained, the out-of-domain system we trained, and the Kaldi-based Samrómur "base" system

|  | WER | OOV-rate total words | OOV-rate unique words |
|---|---|---|---|
| Baseline (Samrómur) | 58.4% | 1.1% | 6.8% |
| Out-of-domain | 36.7% | 0.5% | 3.1% |
| Final | **22.4%** | 0.5% | 3.1% |

Table 2: ASR performance on the Gamli oral history test set

the final models, using the same lexicon and vocabulary.

|  | Out-of-domain LM | Final LM |
|---|---|---|
| Out-of-domain AM | 36.7% | 34.1% |
| Final AM | 24.0% | **22.4%** |

Table 3: ASR performance (WER) on the Gamli oral history test set when combining a specific acoustic model with a specific language model. Note that the final models were trained on the Gamli training set, while the out-of-domain models were not

It seems that acoustic model adaptation had a larger impact than language model adaptation for WER on the Gamli test set.

This is an interesting finding, seeing as language model adaptation is generally more commonly performed, at least in Kaldi, where it takes less computing power than acoustic model adaptation. Though, the results from Table 3 could sim-

ply be due to particularly good out-of-domain text data, they could also suggest that the acoustic elements of oral history are particularly different to other ASR datasets available for Icelandic, and if this is the case, the Gamli training set could be a useful addition to the currently available Icelandic data in order to make acoustic models more robust to elderly speech, historic speech, and historic recording equipment.

# 7 Conclusion and Future Work

In this paper we have presented *Gamli*, a corpus suitable for training speech recognition systems, we have aligned and segmented Icelandic oral histories from manual transcriptions (both OCR from typewritten transcripts and post-edited from ASR output), and filtered out unintelligible segments.

We have described the compilation of the corpus, which has been published under an open license, the origins of the data and evaluation of an ASR system trained on the corpus. We have shown that using the corpus along with other relevant datasets can substantially lower WER for historical speech data, from 58.4% from a baseline system (Samrómur "base" system) to 22.4%. We also draw the conclusion that it could be combined with other ASR training sets which lack in historical recordings and speech from older speakers in order to improve robustness to such audio.

Our final ASR system will be used to automatically transcribe the entire ethnographic audio data stored in Ísmús, i.e. 2,300 hours of audio. We expect the outcome of that process to be in line with the results presented in this paper, with verse, nursery rhymes, singing etc. still remaining a challenge for the customised model, but accuracy for spontaneous speech to be more reliant on audio quality and clarity of speech. Where the quality of these two factors is high, we expect the system to perform well.

Even though the WER may differ substantially for some files, the general outcome will nonetheless be a somewhat readable version of the Ísmús ethnographic collection. As outlined in 1, that output can subsequently be used in a number of ways: first, indexing the Ísmús ethnographic collection for search queries (useful for longer audio files where the description can not do the entire content justice). Second, presenting transcripts alongside the audio as a listening aid and to increase accessibility. Third, as a hypothesis transcript for

post-editing of more transcripts.

The *Gamli* corpus itself should provide an interesting challenge to linguists and ASR researchers interested in spontaneous speech, older speakers, noisy audio, historical recordings and historical dialects.

# 8 Acknowledgements

This paper and the project it is based on are funded by a grant from the Infrastructure Fund from the The Icelandic Centre for Research in collaboration with the Center for Digital Humanities (*Miðstöð stafrænna hugvísinda og lista*) at the University of Iceland. We would like to thank the following people for their collaboration and contribution to the project: Eydís Huld Magnúsdóttir, CEO of Tiro, Rósa Þorsteinsdóttir, curator of the ethnographic collection at the Árni Magnússon Institute for Icelandic Studies and main editor of Ísmús, and Trausti Dagsson, the manager of this project and programmer at the same institute. We would also like to thank our three anonymous reviewers for their comments. Lastly, we thank all the interviewers and interviewees, even though most of them have now gone on to the great beyond, for their invaluable contribution.

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

Steinþór Steingrímsson, Jón Guðnason, Sigrún Helga-
dóttir, and Eiríkur Rögnvaldsson. 2017. Málrómur:
A Manually Verified Corpus of Recorded Icelandic
Speech. In *Proceedings of the 21st Nordic Confer-
ence on Computational Linguistics*, pages 237–240,
Gothenburg, Sweden.

Steinþór Steingrímsson, Sigrún Helgadóttir, Eiríkur
Rögnvaldsson, Starkaður Barkarson, and Jón Guð-
nason. 2018. Risamálheild: A Very Large Ice-
landic Text Corpus. In *Proceedings of the Eleventh
International Conference on Language Resources
and Evaluation*, LREC 2018, pages 4361–4366,
Miyazaki, Japan.

Ravichander Vipperla, Steve Renals, and Joe Frankel.
2008. Longitudinal study of ASR performance on
ageing voices. In *Proceedings of Interspeech 2008*,
pages 2550–2553, Brisbane, Australia.

Rósa Þorsteinsdóttir. 2013. Ísmús (Íslenskur músík- og
menningararfur): An Open-Access Database. *The
Retrospective Methods Network Newsletter*, 7:97–
101.
