# OpenReview forum: "Gamli - Icelandic Oral History Corpus: Design, Collection and Evaluation"
_NoDaLiDa/2023/Conference — NoDaLiDa 2023_

### Official Review · Reviewer_fEqq · 2023-03-10
**Gamli - Icelandic Oral History Corpus: Design, Collection and Evaluation**

**Rating:** 7
**Confidence:** 3

**Review:**

This paper describes the process of making a open-licensed speech corpus for Icelandic oral histories based on historical 20th century recordings from the Department of Ethnology and Folklore at The Árni Magnússon Institute for Icelandic Studies. The paper is well-written and the details of the corpus is presented in a clear and systematic fashion. It is interesting that the authors are able to find a "slight correlation between age and WER". In the evaluation part it would be interesting to see if regional features play any role for the results.

**Paper Type:**

Long paper

---

### Official Review · Reviewer_sj3c · 2023-03-14
**A useful ASR corpus for Icelandic oral histories**

**Rating:** 8
**Confidence:** 3

**Review:**

The paper presents and ASR corpus for Icelandic oral histories. This seems to be a very valuable resource, both in itself and it's potential in speeding up the transcription of more oral history material The description of the process for constructing it is in itself useful for creating simliar resources from similar types of recorded material. I very much would like to see this paper accepted to the conference.

Some comments:
- "Speaker profession is often listed in the metadata, but there is no information about education, and most of the speakers were common people, i.e. workers, farmers, fishermen, housewives etc., with little formal education." I suggest to rephrase this. It sounds a bit condescending. Is it not enough to list the examples of the professions that is given in the metadata. I suggest not to use the phrase "common people", and their education shouldn't matter, they are not hired to analyze the material.
- "The speakers in the collection are from all over the country and therefore reflect the various regional differences in pronunciation". To make the paper more internationally understandable, you could replace "the country" with "Iceland", as people outside of the nordic countries perhaps might not know where Icelandic is spoken.
- "seems to prove well" this sounds like strange English (but I'm not sure)
- "... before aligning the transcripts to the audio and segmenting them ..." This is too short. How did you do the alignment? What tool? How much manual work was required?
- "This step also removes sections with out-of-vocabulary words, which should account for errors stemming from the OCR." This sentence wasn't totally clear. Did you remove parts of the transcriptions with out-of-vocabulary words and the corresponding recorded data? Did you then manually transcribe that part and added it again? If not, would this not possibly introduce a bias in your data (and thereby also in the test data) towards easier material, since out-of-vocabulary words could stem not only from OCR errors, but also that there are rare or dialectal words? Please clarify and perhaps expand. Or is the following paragraph about those segments? In that case, make it clearer.
- I guess "OOV" means out-of-vocabulary? In that case, you could introduce the OOV abbreviation when you mention out-of-vocabulary. If not, you could state what it means.
- "Our final ASR system will be used to automatially transcribe the entire ethnographic audio data stored in Ísmús, i.e. 2,300 hours of audio."
- "That output can subsequently be used in a number of ways: making the data in Ísmús more accessible for the user, both laymen and researchers, indecing the archives for search queries (useful for longer audio files where the description can not do the entire content justice), and as a hypothesis transcriptfor post-editing of more transcripts." It would be good with a some kind of estimation of how useful an ASR system with an error rate of 22% would be for these tasks. Do you have examples of ASR systems previously used for these tasks, their error rate and how useful they were for the tasks?

**Paper Type:**

Long paper

---

### Official Review · Reviewer_gPVh · 2023-03-17
**speech corpus from oral history**

**Rating:** 6
**Confidence:** 3

**Review:**

The paper describes the processing of preexisting data into an ASR corpus for Icelandic.
The paper follows some common processes but might want to cite some of them. For example, the iterative re-training of acoustic models after more material has been aligned has been first outlined for the Spoken Wikipedia Corpora in https://doi.org/10.1007/s10579-017-9410-y (most other steps are also similar to that paper which is why I propose to take a look).
As developed, there is some mismatch between training and test set, given that the latter is likely to not include particular difficulties related to alignment and annotation. It would be interesting to see an error analysis on the test data wrt. to, e.g. normalization issues, from which it would also be possible to derive next steps for improving the corpus/system.
The paper makes some very relevant claims about age distribution and ASR effect. Unfortunately, the small test sets can't prove the claims. All in all, I wonder if something like iVectors could be useful to further improve performance given that speaker characteristics are available for the corpus speakers.

typo: indecing→indexing

**Paper Type:**

Short paper

---

### Decision · Program_Chairs · 2023-03-17

Accept